# Glycoproteomics in Cerebrospinal Fluid Reveals Brain-Specific Glycosylation Changes

**DOI:** 10.3390/ijms24031937

**Published:** 2023-01-18

**Authors:** Melissa Baerenfaenger, Merel A. Post, Pieter Langerhorst, Karin Huijben, Fokje Zijlstra, Joannes F. M. Jacobs, Marcel M. Verbeek, Hans J. C. T. Wessels, Dirk J. Lefeber

**Affiliations:** 1Department of Neurology, Donders Institute for Brain, Cognition, and Behavior, Radboud University Medical Center, 6525 AJ Nijmegen, The Netherlands; 2Division of BioAnalytical Chemistry, AIMMS Amsterdam Institute of Molecular and Life Sciences, Vrije Universiteit Amsterdam, 1081 HZ Amsterdam, The Netherlands; 3Department of Laboratory Medicine, Radboud University Medical Center, 6525 GA Nijmegen, The Netherlands

**Keywords:** cerebrospinal fluid, glycoproteomics, biomarkers, neurodegenerative disease, brain-type glycosylation

## Abstract

The glycosylation of proteins plays an important role in neurological development and disease. Glycoproteomic studies on cerebrospinal fluid (CSF) are a valuable tool to gain insight into brain glycosylation and its changes in disease. However, it is important to consider that most proteins in CSFs originate from the blood and enter the CSF across the blood–CSF barrier, thus not reflecting the glycosylation status of the brain. Here, we apply a glycoproteomics method to human CSF, focusing on differences between brain- and blood-derived proteins. To facilitate the analysis of the glycan site occupancy, we refrain from glycopeptide enrichment. In healthy individuals, we describe the presence of heterogeneous brain-type N-glycans on prostaglandin H2-D isomerase alongside the dominant plasma-type N-glycans for proteins such as transferrin or haptoglobin, showing the tissue specificity of protein glycosylation. We apply our methodology to patients diagnosed with various genetic glycosylation disorders who have neurological impairments. In patients with severe glycosylation alterations, we observe that heavily truncated glycans and a complete loss of glycans are more pronounced in brain-derived proteins. We speculate that a similar effect can be observed in other neurological diseases where a focus on brain-derived proteins in the CSF could be similarly beneficial to gain insight into disease-related changes.

## 1. Introduction

After several advances in the fields of genomics and proteomics over the last few decades, our understanding of the glycoproteome is still lagging behind. This is especially true for the role of glycosylation in the central nervous system and its function in neurological development and disease, where the function of structurally unique “brain-type” glycans is mostly unknown. These brain-type glycans are characterized by the presence of bisecting N-acetylglucosamine, proximal fucose, and truncated antennae on N-glycans [1]. These distinct features make brain-type glycans uniquely different from blood-derived N-glycans. Nevertheless, the diversity of glycan structures, their branching, and the presence of isomeric structures still pose a challenge when it comes to analyzing and understanding the biological function of brain-type glycans or glycoproteins. Although many questions in this area still await an answer, it is known that glycosylation plays a major role in neurological development and function. For central nervous system maturation, the expression of polysialylated structures, especially on the neuronal cell adhesion molecule NCAM, is essential as they strongly contribute to neuronal plasticity and synaptic connectivity [2,3]. However, changes in the expression of glycosyltransferases and changes in protein glycosylation are also observed in neurological diseases associated with ageing such as Alzheimer’s or Parkinson’s disease, highlighting the importance of glycosylation in all stages of neurological development, ageing, and disease [4,5]. Hence, the glycosylation of brain proteins and their changes hold promise for the development of new glycan-based biomarkers and may help to better understand their importance in neurological development and disease.

To understand neurological diseases, cerebrospinal fluid (CSF) can be used, as it reflects the physiological state of the brain more accurately compared to other body fluids such as blood or urine. Several studies using glycomics or individual glycoproteins have shown abnormal glycosylation patterns in the CSF of patients with neurological impairments. For example, glycomics studies in patients with Alzheimer’s disease revealed a decrease in overall sialylation and an increase in bisecting N-acetylglucosamine on N-glycans [6]. Next to released glycans, specific glycoproteins in the CSF show diagnostic potential as well. One of these glycoproteins is transferrin which is present as “brain-type transferrin” and “serum-type transferrin” in the CSF. Here, brain-type transferrin is referring to the proteoforms of transferrin showing brain-type glycans which can be distinguished from serum-type transferrin by the distribution in the number of sialic acids [7]. The ratio of brain-type transferrin to serum-type transferrin in the CSF is altered in neurological diseases such as Alzheimer’s disease and Parkinson’s disease, demonstrating the clinical potential of brain-derived glycoproteins in the CSF [8].

To study the changes in protein glycosylation not only for individual proteins such as transferrin, but holistically for all proteins in a heterogeneous sample such as human CSF, sophisticated technologies are required. Innovations in mass-spectrometry-based glycoproteomics are increasing the identification depth of glycopeptides in complex mixtures together with enhancing the sensitivity, speed, and accuracy of deciphering the glycoproteome of interest. Glycoproteomics technologies have proven to allow the characterization of the glycosylation profile of several hundred proteins simultaneously and can be applied to patient samples such as blood or CSF [5,9].

One major obstacle in the analysis of the CSF glycoproteome is the high abundance of blood-derived proteins. Almost 80% of proteins in the CSF are blood-derived and enter the CSF via the blood–CSF barrier [10]. These blood-derived proteins do not reflect the pathological state of the brain and can overshadow the analysis of brain-derived proteins in glycoproteomics studies. We hypothesized that glycosylation changes in proteins that are solely or predominantly expressed in the brain, such as neuronal cell adhesion molecules or prostaglandin-H2 D-isomerase, can give novel insights into brain-specific disease mechanism and can lead to the development of more accurate and sensitive glycan-based biomarkers in neurological diseases.

To test our hypothesis, we applied state-of-the-art mass spectrometry (MS)-based glycoproteomics to CSF samples from patients diagnosed with congenital disorders of glycosylation, with a range of little-to-severe neurological impairment. To capture complementary information on glycan site occupancy and structural changes in N-glycans, we optimized our workflow and MS acquisition to facilitate high identification rates of glycopeptides in non-enriched samples. Based on gene expression data, we categorized glycoproteins into purely brain-derived proteins and dominantly blood-derived proteins to evaluate if we could detect differences between these two groups. This allowed us to describe the glycoproteome in the CSF of healthy individuals in more detail and to study whether changes in the glycosylation of patients with neurological symptoms are different for proteins originating from the brain or the blood.

## 2. Results

To evaluate the diagnostic potential of CSF glycoproteomics for neurological diseases, we analyzed the CSF from patients with different types of genetic glycosylation disorders (CDGs) and compare it to control samples from healthy individuals. Analyzing the CSF glycoproteome in these patients can give additional insights into the disease mechanism by deciphering the cause of their neurological impairment. We focused on the distinction between brain-derived and blood-derived glycoproteins to highlight tissue-specific differences in glycosylation. We chose to not perform glycopeptide enrichment to avoid the loss of highly truncated and missing glycans, which are important features in some CDG patients.

### 2.1. Glycoproteomics Identification in Non-Enriched CSF Samples

To characterize the glycoproteome in the CSF from patients diagnosed with CDG and in healthy controls, we performed LC-MS/MS measurements on non-enriched tryptic glycopeptides. However, several challenges arise when working with non-enriched samples due to the low overall protein concentration and the vast dynamic range of protein abundance in the CSF [10].

To overcome these disadvantages, we performed as few steps as possible during sample preparation and used highly sensitive nano-LC separation coupled to a high-resolution ESI-QqTOF-MS instrument (timsTOF Pro, Bruker) to generate MS/MS spectra for peptide and glycopeptide identification (Figure 1A). We leveraged the high sensitivity and fragmentation scan speed of parallel accumulation–serial fragmentation (PASEF) for data-dependent acquisition to identify glycopeptides in non-enriched heterogeneous CSF samples [11,12]. Additionally, we excluded singly charged precursor ions as singly charged peptides and glycopeptides usually result in poor CID fragmentation spectra.

The glycopeptides were identified using MSFraggerGlyco. After identification, the relative abundances of glycopeptides were obtained by peak integration using Skyline [13,14]. The combined MSFraggerGlyco output file and the results after peak integration with Skyline can be found in Appendix A, respectively. In total, we were able to identify over 800 unique glycopeptides from almost 200 different N-glycosites in 25 CSF samples without using glycopeptide enrichment. The number of unique glycopeptides also includes glycopeptides with a complete loss of glycosylation (Figure 1B). Thus, we identified about one-third of the number of unique N-glycopeptides reported in the CSF in other glycoproteomics studies in which glycopeptide enrichment was performed [9]. However, we were able to detect glycopeptides with highly truncated glycans and non-glycosylated peptide variants showing a lack of glycosylation, which can be used to evaluate differences in glycan occupancy and can be of diagnostic value.

Within our set of identified glycopeptides, around 20% originate from brain-derived proteins based on expression data from the Human Proteome Atlas, which is in good agreement with the reported number of brain-derived proteins and blood-derived proteins in the CSF [10,15,16]. Although this classification based on expression data is not possible for all proteins due to expressions in multiple tissues, it enabled us to focus on brain-specific glycosylation patterns. Figure 1B,C show the detected number of unique N-glycopeptides, corresponding N-glycosites, and glycoproteins, as well as the distribution of different glycan structures per glycopeptide divided into brain-derived and blood-derived proteins. The reoccurring ratio of 80% blood-derived protein and 20% brain-derived proteins is evident and can be observed not only on the glycopeptide level but also when looking at the number of corresponding glycoproteins. The dominant brain-derived protein in the CSF is prostaglandin-H2 D-isomerase (PTGDS), for which we detected more than 50 different unique glycopeptides for two N-glycosites within our cohort. Similarly heterogenous is the mostly blood-derived protein alpha-1-antitrypsin (A1AT), for which we detected 61 unique N-glycopeptides over 3 N-glycosites.

### 2.2. Characteristics of the CSF Glycoproteome in Healthy Individuals

It has been known for several decades that glycosylation in the nervous system shows a clear distinction from glycosylation in other tissues [17]. So-called brain-type glycans can be found on glycoproteins and other glycoconjugates that are characterized by the presence of bisecting N-acetylglucosamine, proximal fucose, and truncated antennae on N-type glycans [1]. To characterize the CSF glycoproteome, we first focused on determining the glycosylation signatures in healthy individuals. Using our classification of brain-derived proteins and blood-derived proteins, we visualized the dominant glycan compositions including the relative protein contribution for each glycan (Figure 2). Based on the glycan composition, we proposed potential glycan structures, as shown in Figure 2 and the subsequent figures. Other glycan structures with the same composition are equally plausible and cannot be unambiguously distinguished by our workflow. With this, we were able to recreate typical plasma protein N-glycosylation for blood-derived proteins (Figure 2, left), which is in good agreement with the overview on plasma protein glycosylation proposed by Clerc et al. [18]. The dominant glycan composition corresponds to a biantennary, fully sialylated glycan without fucose and bisecting N-acetylglucosamine, which is detected on proteins such as transferrin, hemopexin, and alpha-1-antitrypsin. Highly sialylated tri- and tetra-antennary glycans were observed for alpha-acid glycoproteins 1 and 2, and immunoglobulin IgG1 showed typical glycan compositions such as H3N4F1S0 (G0F) and H4N4F1S0 (G1F).

The relative protein contribution to each glycan composition for brain-derived proteins is, as expected, dominated by prostaglandin-H2 D-isomerase in protein abundance as well as glycan heterogeneity. Interestingly, we detected the presence of high-mannose glycans for the neuronal cell adhesion molecule (NCAM, pink bar). This protein plays a key role in early neuronal development, where the expression of polysialylated glycan structures leads to anti-adhesive properties and thus defines nervous system plasticity [19]. However, postnatal development leads to the downregulation of the polysialylated NCAM and an increase in the nonpolysialylated NCAM with increasing age [20]. Here, we could detect the presence of high-mannose glycan structures on the NCAM which has been reported before for the bovine NCAM [21,22]. Additionally, we detected brain-type N-glycans for several glycosylation sites of the NCAM.

Even though our classification of proteins based on expression data reflects serum-type glycosylation and brain-type glycosylation, many proteins that are highly abundant in the blood are also expressed in the brain. This has been extensively described for transferrin and the term brain-type transferrin was established for the non-sialylated proteoforms [1,23]. Since then, brain-type transferrin was proposed to be diagnostic for many neurological diseases such as Alzheimer’s disease, vanishing white matter disease, or spontaneous intracranial hypotension [7,8,24,25,26]. Our data additionally suggest that transferrin not only shows brain-type proteoforms but other proteins as well. Figure 3 shows the relative abundance of glycopeptides for one selected N-glycosite of four different proteins. Their relative intensities were obtained after the integration of the corresponding extracted ion chromatograms and normalization per N-glycosite. The intensity of individual glycan structures represents the average abundance in the control population. Glycans with two or three brain-type features, namely fucosylation, bisecting N-acetylglucosamine, and truncated antenna, were classified as brain-type glycans, whereas glycans with only one or no brain-type feature were classified as serum-type glycans.

Dickkopf-related protein 3 (DKK3) is a protein highly expressed in the brain and spinal cord. Glycans expressed for N-glycosite N204 (peptide sequence: GSNGTICDNQR) only show brain-type glycans exhibiting motifs such as fucosylated glycans and bisecting N-acetylglucosamine (Figure 3, top). For clusterin (site N291, peptide sequence HNSTGCLR), which is expressed in multiple tissues such as liver and brain, we were able to detect glycans showing brain-type features but also glycans corresponding to serum-type glycans in almost equal amounts. For transferrin (site N630, peptide sequence: QQQHLFGSNVTDCSGNFCLFR), we detected fewer brain-type glycans and no brain-type glycans for apolipoprotein D (site N65, peptide sequence: CIQANYSLMENGK). As mentioned above, about 80% of the proteins in the CSF are predominantly blood-derived, but only rarely are blood-specific and expressed in the brain as well. Both brain-derived and blood-derived fractions contribute to the protein concentration in the CSF. We speculated that the mixed glycosylation pattern present on a glycoprotein relates to the relative proportion of protein expressed by the brain vs. protein coming from the bloodstream and being transported to the CSF via the blood–CSF barrier. This has been demonstrated before for transferrin and is supported by the data shown in Figure 3 [7,8]. Additionally, we proved that glycoproteins such as dickkopf-related protein 3, which are only expressed in the brain, show brain-type glycosylation, whereas other proteins such as clusterin, which are expressed in multiple tissues, also show this in their glycosylation pattern.

### 2.3. Glycosylation Changes in CDG Patients

Brain-derived proteins show a great promise to be used as potential biomarkers for several neurological diseases. However, most glycoproteomic approaches in the CSF do not distinguish between brain-derived proteins and non-brain-derived proteins, watering down the diagnostic potential of CSF glycoproteomics. To investigate the potential benefit of distinguishing between brain-derived and non-brain derived proteins, we analyzed the CSF glycoproteome from patients diagnosed with different types of CDG, 1 patient suffering from alcohol abuse, 2 patients without diagnoses showing neurological symptoms, and 11 healthy controls.

CDG patients suffer from genetic defects that affect the cellular glycosylation machinery, which can lead to the altered glycosylation of proteins, lipids, or other biomolecules. Since several hundred enzymes such as glycosyltransferases, isomerases, or membrane transporter proteins are involved in the process of glycosylation, a plethora of different mutations can lead to a CDG. Even though the clinical phenotype varies, and symptoms can show differences in severity depending on the exact mutation, most CDGs are accompanied by neurological manifestations. However, research and diagnostics in relation to CDGs are mostly performed on plasma proteins, e.g., by isoelectric focusing of transferrin or apolipoprotein CIII isoforms [27]. Here, we characterized the brain glycoproteomes of several CDG patients to understand the neurological symptoms observed in most CDG patients. A schematic overview of the affected genes and their role in the glycosylation machinery for patients in this study is shown in Figure 4.

We included genetic defects that are involved in sugar donor generation such as MPI, PMM2, or NANS, as well as defects of the endoplasmic reticulum affecting the early steps of N-glycosylation (ALG1, ALG6, SRD5A3, DMP1, and DPM3) and defects of the Golgi (SLC35A1 and ATP6V0A2). We included one patient suffering from alcohol abuse for which we expected stronger changes in glycosylation among blood-derived proteins as well as two patients without a clear diagnosis who suffer from neurological symptoms but are most likely not CDG patients. An overview of the clinical symptoms and patient characteristics can be found in Appendix A. We hypothesized that CDG cases with a dominant neurological phenotype will show stronger changes in glycosylation on brain-derived proteins as compared to non-brain-derived proteins.

It is known that both CDG and alcohol abuse can lead to a loss of glycosylation. This loss of glycosylation can be detected by measuring carbohydrate-deficient transferrin [28,29,30]. For these, but also for other diseases and applications, it is relevant to not only consider changes in glycan structures but also N-glycosite occupancy, similar to our approach. In our cohort, we were able to detect 31 unique glycopeptides with a complete loss of glycosylation. These non-glycosylated glycopeptides were observed in transferrin as well as other glycoproteins (such as alpha-1-acid-glycoprotein 1 or 2; ceruloplasmin; clusterin; and complements C3, C5, and C9). Furthermore, purely brain-derived proteins such as prostaglandin-H2 D-isomerase or contactin-1 also showed a complete loss of glycosylation for specific glycosites and patient samples. A complete overview of detected glycopeptides and the relative abundances of glycopeptides including glycopeptides with a complete loss of glycosylation can be found in Appendix A.

To compare the overall changes in glycosylation on brain-derived and non-brain-derived proteins, we grouped glycans based on their glycan composition into glycan classes and considered the relative abundance of all detected glycopeptides for all proteins. We distinguished between a complete loss of glycans and highly truncated glycans if a classical differentiation between complex-type, hybrid, or high-mannose glycan was not possible. Secondly, we were interested in brain-type glycan features including short antennae with a lack of sialic acid, bisecting N-acetylglucosamine, and fucosylation.

Figure 5 shows the glycan class representation for brain-derived and non-brain-derived proteins. The results for 11 healthy controls were plotted as a violin plot to represent the distribution in the control. Results from patient samples were overlaid and single data point per patient. We did not group different CDG types as the glycosylation varied tremendously between different patients according to affected genes. Additionally, even within CDG types, a low correlation between the mutant alleles and phenotypes has been observed and disease severity has been found to depend on zygosity, residual enzyme activity, and interventive nutritional treatment [31,32,33].

In the control population, as expected, we observed a high degree of fucosylation and bisecting N-acetylglucosamine and a low degree of sialylation on brain-derived proteins. Interestingly, the amount of truncated glycans appeared to be higher on brain-derived proteins, varying from 0% to 4.5% relative abundance levels in healthy individuals. For CDG patients suffering from NANS-CDG, ATP6V0A2-CDG, or SRD5A3, we observed only marginal differences in glycosylation compared to the control population. In contrast, for patients suffering from ALG1-CDG, ALG6-CDG, and PMM2-CDG, a tremendous change in glycosylation was evident. For the patient diagnosed with ALG6-CDG, we again observed a stronger loss of glycosylation for brain-derived proteins compared to non-brain-derived proteins. In addition, we observed an increased expression of high-mannose glycans exclusively for brain-derived proteins for the patients diagnosed with ALG6-CDG.

The literature shows that patients with CDG-type I, such as ALG6 or ALG1, show an increase in fucosylated glycans on plasma proteins [34]. This trend is also observed in our data for blood-derived proteins for patients diagnosed with ALG6 and ALG1-CDG. However, brain-derived proteins show the opposite trend with a decrease in fucosylation.

Figure 5 also shows other patients with abnormal glycosylation. The patient diagnosed with SLC35A1 displays a strong reduction in sialylation on non-brain-derived proteins. This is consistent with the defect, as SLC35A1 encodes a nucleotide sugar transport protein that enables the transportation of CMP-sialic acid into the Golgi, where it is used as a glycosyl donor for sialylation. Reduced sialylation has been described previously for blood proteins in patients diagnosed with SLC35A1 [30,35,36]. Here, we confirmed that this effect is less pronounced for brain-derived proteins (Figure 5), possibly because reduced sialylation is characteristic of baseline brain-type glycosylation.

As expected, we also observed a similar trend for the patient diagnosed with alcohol abuse. Here, non-brain-derived proteins show a higher amount of truncated glycan structures compared to brain-derived proteins. It is thought that ethanol and its metabolite acetaldehyde can cause Golgi remodeling and have inhibitory effects on glycosyltransferases [37,38,39]. For example, acetaldehyde reduces the activity of the phosphomannomutase PMM2, which can cause the failure of complete glycan maturation or disrupt the en bloc transfers of the oligosaccharide precursor to the asparagine side chain of the nascent polypeptide in the ER (see Figure 4) [40]. We hypothesized that this effect is more pronounced in the liver, the central location for alcohol metabolism, which can explain the strong changes in glycosylation observed for blood proteins compared to little changes on brain-derived proteins.

### 2.4. Severe Changes in Glycosylation for Patient ALG6 and PMM2

Within our CDG cohort, some patients show a dramatic change in glycosylation including a loss of glycosylation, whereas other cases only show minor or no differences. The patient with a genetic defect in ALG6 shows strong glycosylation abnormalities, as well as severe clinical symptoms such as mental disorders, motoric retardation, seizures, and epilepsy. The glycosylation abnormalities can be found on both brain-derived and non-brain-derived proteins, as shown in Figure 6, exemplary for hemopexin and prostaglandin-H2 D-isomerase. For hemopexin, glycosylation site N187, we dominantly detected one glycan in healthy controls corresponding to a biantennary, fully sialylated complex-type glycan. The patient diagnosed with ALG6-CDG showed a significant shift in glycan stoichiometry and a strong decrease in glycan occupancy. Similar effects were also seen for patient PMM2_a.

Interestingly, we detected highly truncated glycans, e.g., the attachment of only one N-acetylglucosamine or the N-tetrasaccharide H1N2F0S1. This N-tetrasaccharide has been described previously for patients with ALG1-CDG, ALG2-CDG, PMM2-CDG, and MPI-CDG, either on intact transferrin or with total plasma glycomics, and was proposed as a specific biomarker for these CDG subtypes [41]. Our results link these unusual glycan structures (tetrasaccharide H1N2F0S1 and fucosylated pentasaccharide H1N2F1S1) to several proteins, including alpha-1-antitrypsin, alpha-1-antichymotrypsin, alpha-2-macroglobulin, and brain-derived prostaglandin-HD 2-isomerase. We detected these glycan structures for PMM2-CDG, ALG1-CDG, and ALG6-CDG and in one patient without diagnosis (no diagnosis I) in small amounts. It is hypothesized that the reduced mannosylation of lipid-linked chitobiose in the ER leads to the attachment of chitobiose to the nascent protein. The chitobiose modification can be processed further in the Golgi to produce the N-tetrasaccharide and its fucosylated form [41,42]. However, detecting the N-tetrasaccharide in the patients diagnosed with ALG6-CDG suggests a more complex mechanism as the mutation would not directly link to a reduced mannosylation in the ER (see Figure 4). Additionally, we detected the modification of glycopeptides with only one N-acetylglucosamine, which can also not be explained by reduced mannosylation in the ER. The patients diagnosed with ALG6-CDG and PMM2-CDG show a strong abnormal glycosylation pattern for brain-derived prostaglandin-D2 H-isomerase (Figure 6). We detected the N-tetrasaccharide in small amounts and several other truncated glycans which we expected to be pauci-mannose glycans. Remarkably, the glycosylation pattern strongly deviates from the control, showing a tremendous amount of complete loss of glycosylation as well as a modification with only one N-acetylglucosamine. We would like to acknowledge that these relative proportions of glycopeptide abundance should be regarded with caution as non-glycosylated peptides often show better ionization properties compared to glycosylated peptides [43]. Nevertheless, we did observe a significant signal increase in glycopeptides with short or absent glycans for the patient diagnosed with ALG6-CDG and PMM2-CDG, which is more pronounced on brain-derived prostaglandin-H2 D-isomerase than on hemopexin. Overall, we detected several glycosylation characteristics that are differently pronounced on brain-derived vs. blood-derived proteins in different CDG patients such as ALG6-CDG, PMM2-CDG, or SLC35A1-CDG. These data imply that considering tissue-specific glycosylation can have a profound impact on detecting differences in disease cohorts.

## 3. Discussion

Here, we show that glycoproteomics can be performed in non-enriched CSF samples, with the advantage that peptides with a complete loss of glycosylation or glycopeptides with highly truncated glycans can be detected. We could therefore estimate N-glycosite occupancy, which is of utmost importance for characterizing CDG cases, but also in monitoring other diseases where glycan truncation plays a role, such as alcohol abuse or other infectious, genetic, or metabolic diseases [37,44,45]. Furthermore, this aspect is completely overlooked in current glycomics studies on neurodegenerative diseases.

Performing glycoproteomics studies on brain-derived proteins, which account for only about 20% of proteins in the CSF, and non-enriching glycopeptides can compromise identification depth due to their relative low abundance. The development of PASEF acquisition already demonstrated its use for increased identification depth for proteomics applications. Using this technology enabled us to perform glycoproteomics in complex non-enriched CSF samples.

By including the expression data of the discovered glycoproteins, we could distinguish between proteins that are predominantly brain-derived and those that are not. This classification should, however, be regarded with caution as many proteins are expressed in multiple tissue types, complicating a clear distinction. However, our classification reveals a typical serum-type glycosylation pattern for non-brain-derived proteins and a hugely different overall glycosylation pattern for brain-derived proteins, which have mainly brain-type glycans with truncated antennae, bisected N-acetylglucosamine, and a higher degree of fucosylation. We hypothesized that distinguishing between brain-derived and non-brain-derived proteins is beneficial in the study of neurological diseases and demonstrated this for patients diagnosed with different types of CDG. We proved that, in some CDG cases, e.g., ALG6-CDG, the complete loss of glycans is more pronounced in brain-derived proteins compared to plasma-derived proteins. In patients diagnosed with alcohol abuse, the opposite is true. Here, non-brain-derived proteins are more affected and have shortened or absent glycans. However, our relative quantification of glycopeptides, highly truncated glycopeptides, and glycopeptides with absent glycans has to be regarded with caution since the size and composition of the glycan can influence ionization. Hence, we suspected an overrepresentation of glycopeptides with highly truncated glycans and a complete loss of glycosylation in our data. Additionally, we acknowledged that our sample size for different CDG defects is small and our observations should be regarded with caution. The different genetic defects presented here led to diverse alterations in glycosylation and the comparisons within our CDG cohort represent challenges due to their clinical and genetic features. Moreover, the data from a single patient with a history of alcohol abuse might not be representative of a larger cohort and should only be regarded conceptually. We included this patient in contrast to the CDG cohort to demonstrate that glycosylation abnormalities can also be more pronounced on blood-derived proteins. Consequently, further studies are required to unravel disease mechanisms on a glycoproteomics level in CDG patients but also other patient groups. However, the work presented here lays the foundation for studying the role of brain-derived glycoprotein alterations in CDG diseases and beyond.

Since the majority of proteins in the CSF do not originate from the brain but from the liver, plasma cells, or other tissues, results from these proteins could overshadow brain-specific changes in other glycoproteomics approaches. We expected that focusing on proteins that originate from the brain would have a positive impact on glycoproteomics studies in other neurological diseases such as Alzheimer’s disease, where the brain is the center of the disease. Here, brain-derived glycoproteins, such as prostaglandin-H2 D-isomerase, have shown changes in protein abundance in a variety of different neurological diseases [46,47,48,49]. Additionally, glycosylation can also influence protein function in neurological diseases. For example, the glycosylation of clusterin regulates the neurotoxicity of amyloid-beta peptides in Alzheimer’s disease where non-glycosylated clusterin can provide a partial protection [50]. The approach demonstrated here allows such effects to be investigated further by describing the glycoproteome, including the absence of glycosylation, in patient material. With this, our methodology will help to clarify the involvement of brain glycosylation and its changes in neurological diseases.

## 4. Materials and Methods

### 4.1. CSF Samples

The patient and control samples were collected at the Radboudumc in accordance with the Declaration of Helsinki and CMO approval 2019-5591 for biomarker studies in diagnostic samples of CDG patients and controls. An overview of the clinical information of and sampling date of CDG patients can be found in Appendix A. CSF control samples were left-over samples from diagnostic investigations, from patients who were suspected of neurological disease, but were found to be free of neurological disease. These CSF samples were provided anonymously.

### 4.2. Digestion of Proteins and Glycoproteins from CSF

Glycoproteins and proteins from the 100 µL CSF sample were precipitated by adding 300 µL of cold ethanol. Precipitation was completed overnight at −80 °C. The protein pellet was centrifuged for 15 min at 14,000 rpm and freed from the supernatant. Proteins were resuspended in 7.5 µL of 8 M urea and denatured by adding 7.5 µL of 100 mM DTT and initiating incubation at room temperature for 30 min. Alkylation was completed by adding 7.5 µL of 50 mM 2-chloroacetamide in 50 mM ammonium bicarbonate buffer and incubation at room temperature in the dark for another 30 min. The sample solution was diluted with 15 µL of 50 mM ammonium bicarbonate buffer and 2 µL of protease Lys-C (mass spectrometry grade, Wako Chemicals, Huissen, The Netherlands) was added. The digest was performed for 3 h at room temperature. Subsequently, 100 µL of 50 mM ammonium bicarbonate buffer and 2 µL (12.5 ng/μL sequencing-grade modified trypsin, Promega, Leiden, Netherlands) of trypsin were added. The tryptic digest was performed overnight at 37 °C. The trypsin was inactivated by heating the sample solution to 95 °C for 5 min.

### 4.3. LC-MS/MS Measurements

Non-enriched peptides and glycopeptides were directly analyzed by LC-MS/MS after tryptic digest in duplicate measurements. For that, 2 µL of the sample solution was injected on a nano-LC system (nanoElute, Bruker, Bremen, Germany), equipped with a reversed-phased C18 column (0.075 × 150 mm, 1.9 um particle size, Bruker fifteen, Bruker). LC separation was completed at 45 °C with a flow rate of 0.5 µL/min and a mobile phase of solvent A (0.1% formic acid (FA) and 0.01% trifluoroacetic acid (TFA) in water) and solvent B (0.1% FA and 0.01% TFA in acetonitrile). The gradient was set as follows: 0 min, 3% B; 60 min, 42% B, 62 min, 80% B; and 67 min, 80% B. Online MS measurements were performed on a trapped ion mobility spectrometry quadrupole time-of-flight mass spectrometer with an electrospray ionization source (ESI-TIMS-QqTOF, timsTOF Pro, Bruker Daltonik, Bremen, Germany) in the positive ion mode. Prior to sample injection, mass and mobility calibration was performed using sodium formate clusters. We increased the ionization efficiency and the charge states of glycopeptides by using dopant-enriched nitrogen gas as source gas by means of a nanobooster, as described previously [51,52]. For this, a captive spray nanoflow source was used, equipped with a nanoBooster (Bruker Daltonik) and acetonitrile as the source gas dopant, set to a pressure of 0.20 bar. The source-drying temperature was set to 180 °C. To reduce in-time fragmentation of the glycopeptide precursor, delta potentials were slightly adapted compared to standard proteomics settings. Delta potentials were set as follows: D1 = −20 V, D2 = −150 V, D3 = 110 V, D4 = 110 V, D5 = 0 V, and D6 = 55 V. MS and MS/MS spectra were acquired between a mass range of 100 *m*/*z* and 4000 *m*/*z* with an acquisition rate of 10 Hz. We excluded singly charged precursor ions by selecting a two-dimensional region (*m*/*z* and mobility). Ions with a charge state of +2 and higher were selected for parallel accumulation–serial fragmentation (PASEF) acquisition by applying a polygon filter in *m*/*z* and mobility dimension excluding singly charged background ions. PASEF-MS/MS scans were acquired until a target intensity of 100,000 a.u. was reached with a total cycle time of 1.80 s. Linear collision energy stepping as a function of increased mobility from 0.6 to 2.0 1/K_0_ was used to acquire high collision energy spectra for peptide fragmentation (65 eV to 100 eV) and low collision energy spectra for glycan fragmentation (45 eV to 65 eV). For the first step, a collision RF of 1600 V, a transfer time of 100 us, and a pre-pulse storage of 13 us were used for spectrum acquisition. For the second step, the collision RF was set to 1600 V, the transfer time was set to 80 us, and the pre-pulse storage was set to 10 us. The LC-MS/MS data were deposited to the ProteomeXchange Consortium with the dataset identifier PXD038787 and can be reviewed with the following reviewer account details: Username: reviewer_pxd038787@ebi.ac.uk; Password: waajM1vG. All spectra were uploaded as an mzML file and CSF samples can be identified with the provided file *Sample information.xlsx*. The MS/MS spectra used for glycopeptide identification can be retrieved based on the exact mass, charge, retention time, and ion mobility provided in Appendix A.

### 4.4. Identification of Glycopeptide Spectra and Target List Generation

Glycopeptides were identified using MSFragger-Glyco (version MSFragger-3.1.1) with Fragpipe 15.0 and philosopher (version 3.6.0.) [13]. A higher-energy collision dissociation (HCD) offset search was used to identify glycopeptides based on glycan mass offsets and peptide backbone fragments. Diagnostic B- and Y-ions from glycan fragmentation were specified to identify the glycopeptide spectra, and the glycan mass offsets of potential N-glycan masses were defined. A list of used glycan mass offsets and diagnostic fragment ions can be found in the Appendix A (fragger.params). Glycan mass offsets were based on common human N-glycan masses as well as highly truncated glycans often seen in severe cases of CDG [35,41,42,53]. For the identification of the peptide backbone, the reviewed human database (including contaminants and decoys) was used (Uniprot ID UP000005640, downloaded 7 April 2021). A mass tolerance of ±20 ppm was applied, the carbamidomethylation of cysteine was defined as fixed modification and methionine oxidation, and pyroglutamic acid formation from N-terminal glutamine and N-terminal acetylation were specified as variable modifications. Further settings for glycopeptide identification can be found in the Appendix A (fragger.params and fragpipe_2021-07-13_14-45-01.config).

The protein sequence matches (psm.tsv) from MSFragger-Glyco were additionally filtered and only identifications with a peptide prophet probability of >0.95 and a hyperscore > 15 were accepted. The chemical composition of accepted glycopeptides was calculated and glycopeptides with unique peptide moieties, glycan moieties, and other modifications and charges were used to compile a target list for target extraction. The retention time and mobility of glycopeptides with multiple spectral identifications were both averaged.

### 4.5. Target Extraction and Relative Quantification of Glycopeptides

The extracted ion chromatograms of glycopeptides and peptides with a complete loss of glycosylation were created and integrated using Skyline (version 21.1.0.278) for both duplicate injections. Glycopeptides identified from MSFragger-Glyco were matched with peptide identification from label-free quantification from MSFragger to detect the complete losses of glycosylation of identified glycopeptides. Glycopeptides and peptides with N-glycosylation sites showing a complete loss of glycosylation were combined in one target list. Glycopeptide and peptide masses were extracted for the first four isotopic peaks with a mass window of 0.015 Da, a resolving power of 50,000, a mobility window of 0.2 1/K_0_, and a retention time window of 2 min. Retention time windows for integration were manually checked and adapted when necessary.

Integrated peaks with a peak area below 10.000 were discarded and peak areas for the same glycopeptides but different isotopic peaks, charge states, and modifications were summed. This way, integration results for unique glycopeptides were obtained. Integration results were either normalized based on the total intensity to obtain a relative quantification of all detected glycopeptides or a normalization procedure based on the total intensity per glycosite was completed to follow the glycosylation of individual proteins [54].

## 5. Conclusions

CSF is an ideal source for biomarker development and the study of neurological diseases because it is in direct contact with the brain. However, it is important to consider that most proteins originate from the blood and only about 20% of proteins originate from the drainage of the interstitial fluid of the central nervous system. We used a glycoproteomic approach in the CSF where we examined changes in glycosylation and site occupancy while distinguishing proteins into brain-derived and non-brain-derived proteins based on expression information. We demonstrated that some CDG patients show stronger glycosylation abnormalities on brain-derived proteins and hypothesize that focusing on brain-derived proteins will not only reveal stronger glycosylation changes in CDG patients with neurological manifestation, but will also be of advantage for research and diagnostic of other neurological diseases.

## Figures and Tables

**Figure 1 ijms-24-01937-f001:**
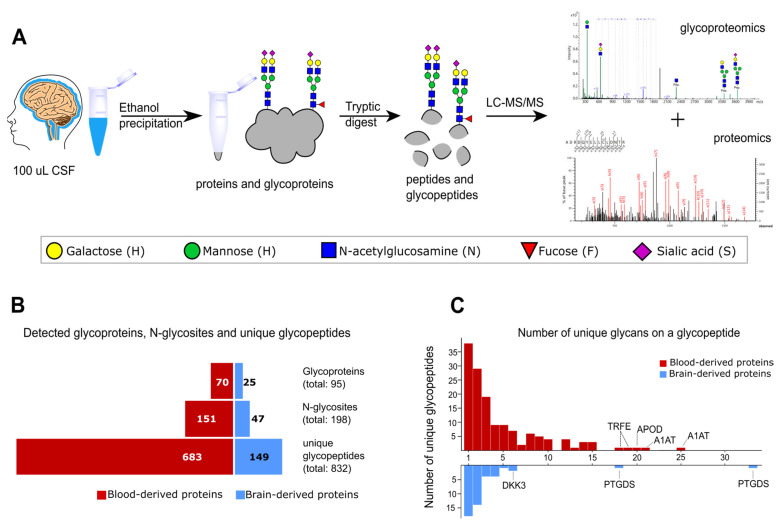
Glycoproteomics workflow. (**A**) In total, 100 µL of CSF was collected from CDG patients and healthy controls. Proteins and glycoproteins were enriched by protein precipitation with ice-cold ethanol. After resuspension and tryptic digest, peptides and glycopeptides were analyzed by LC-MS/MS. The glycoproteomics data were interpreted using MSFragger Glyco. (**B**) Although our workflow does not include any glycopeptide enrichment, we were able to detect over 800 unique glycopeptides from 198 different N-glycosites originating from 95 different glycoproteins. Around 20% of detected unique glycopeptides, N-glycosites, and glycoproteins originate from purely brain-derived proteins. (**C**) Glycosite heterogeneity on detected N-glycosites is visualized for blood-derived and brain-derived proteins. For most N-glycosites, one or two unique glycans were detected. Some glycosites show a variety of different glycan structures, as seen on N-glycosylation sites of transferrin (TRFE), alpha-1-antitrypsin (A1AT), apolipoprotein D (APOD) or brain-derived prostaglandin-H2 D-isomerase (PTGDS), and dickkopf-related protein 3 (DKK3).

**Figure 2 ijms-24-01937-f002:**
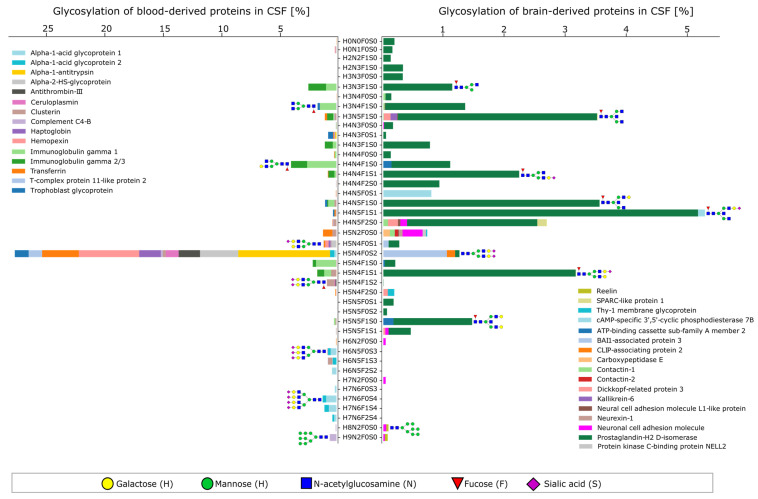
Glycan overview for the most abundant glycan compositions and glycoproteins. Based on the relative intensity of glycopeptides, a relative protein contribution for each glycan is represented. Blood-derived proteins show a characteristic serum-like profile of glycans with mainly fully sialylated complex type N-glycans. Brain-derived proteins show a higher degree of fucosylation, bisecting N-acetylglucosamine, and truncated antennae. Exemplary annotated glycan structures represent one possible translation of the glycan composition.

**Figure 3 ijms-24-01937-f003:**
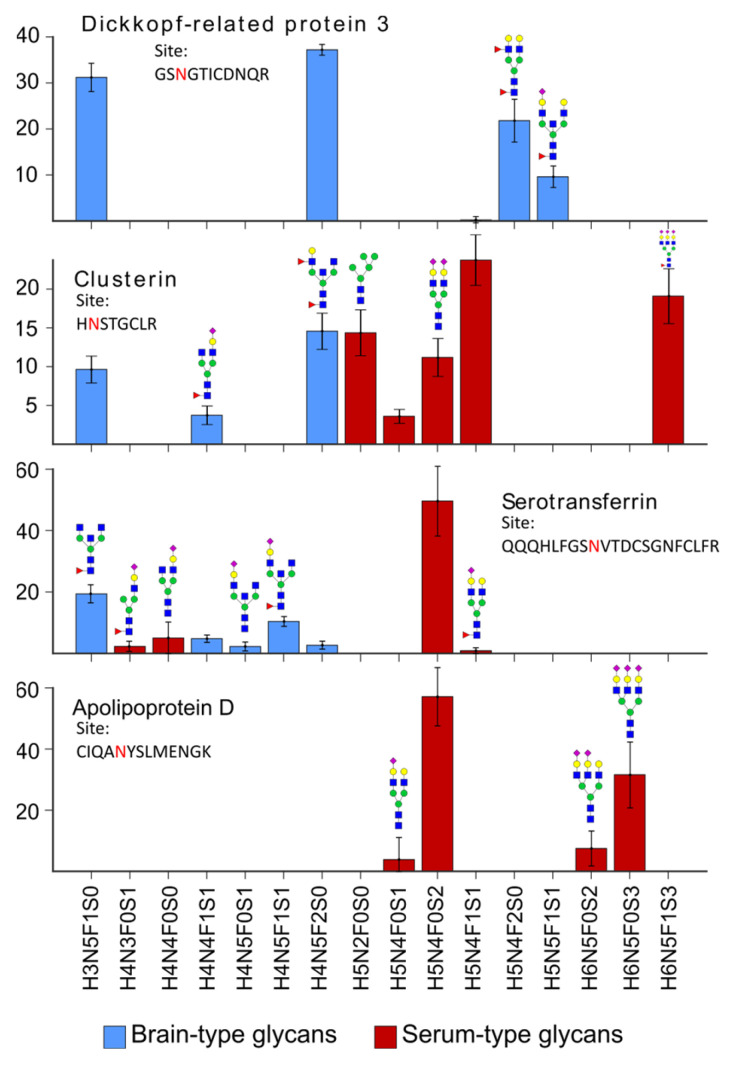
Glycopeptide profiles as average over eleven control samples for four different glycoproteins. Glycans with at least two brain-type features (fucosylation, bisecting N-acetylglucosamine, and truncated antenna) are colored in blue and glycans with blood-type features are colored in red. Purely brain-derived proteins such as dickkopf-related protein 3 mainly show brain-type glycans. Proteins which are expressed in multiple tissues, such as transferrin or clusterin, show glycan structures which are commonly found on blood proteins but also brain-type glycans in lower abundance. Apolipoprotein D shows only blood-type glycans, which can be explained by its low expression in brain tissue.

**Figure 4 ijms-24-01937-f004:**
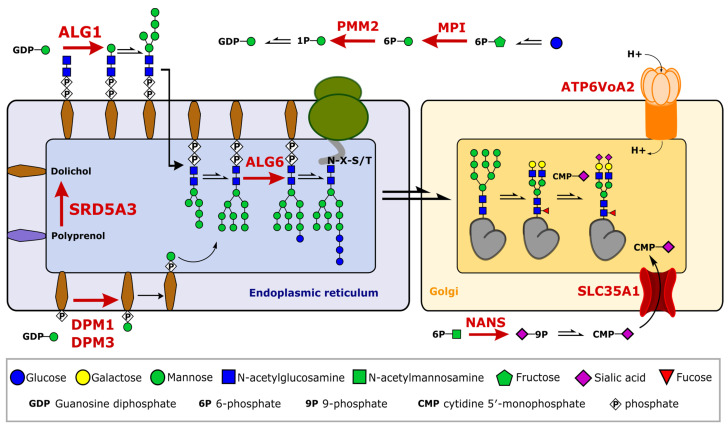
Simplified protein N-glycosylation pathways observed in endoplasmic reticulum (ER) and Golgi. Affected genes of CDG patients and corresponding enzymatic functions are shown in red. Sugar phosphates are obtained by converting monosaccharides such as glucose to phosphorylated sugars such as mannose-1-phosphate (1P) catalyzed by enzymes such as mannose-6-phosphate isomerase (MPI) or phosphomannomutase 2 (PMM2). Nucleotide sugars such as GPD-mannose or CMP-sialic acid are used as glycosyl donors in protein glycosylation. Transporter proteins including SLC35A1 and changes in Golgi pH caused by genetic defects in ATP6V0A2 (ATPase H^+^ transporting V0 subunit a2) influence protein glycosylation too.

**Figure 5 ijms-24-01937-f005:**
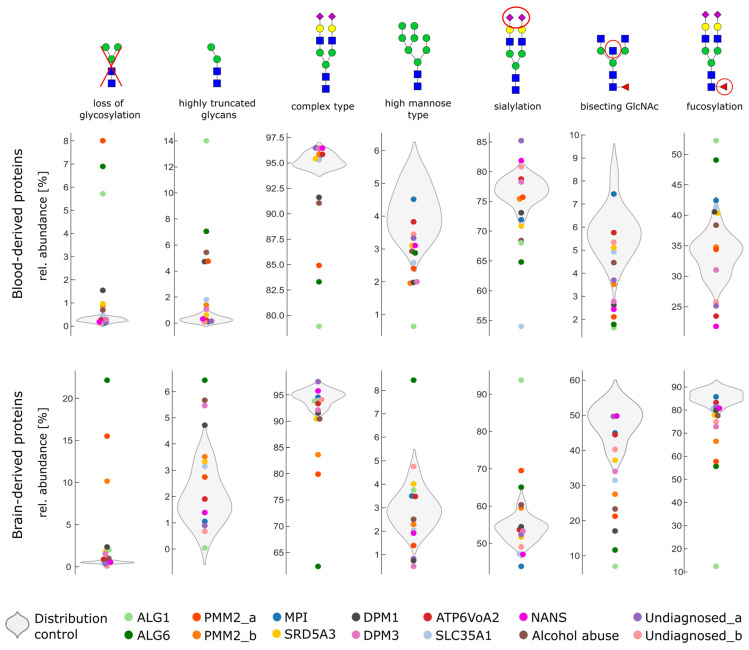
Glycan class variability of CDG patients, two patients without diagnoses, and one patient diagnosed with alcohol abuse compared to the glycan distribution in controls (grey violin). Severe changes in glycosylation are observed in the patients diagnosed with ALG1-CDG and ALG6-CDG, as well as both PMM2-CDG patients. Here, the complete loss of glycosylation is more pronounced on brain-derived proteins. As expected, the patient diagnosed with alcohol abuse shows a higher level of truncated glycans on blood-derived proteins compared to brain-derived proteins. The patient diagnosed with SLC35A1 (the impairment of CMP-sialic acid transporter) shows lower levels of sialylation for blood-derived proteins and almost normal levels for brain-derived proteins, potentially due to lower levels of sialylation seen on brain-type glycans.

**Figure 6 ijms-24-01937-f006:**
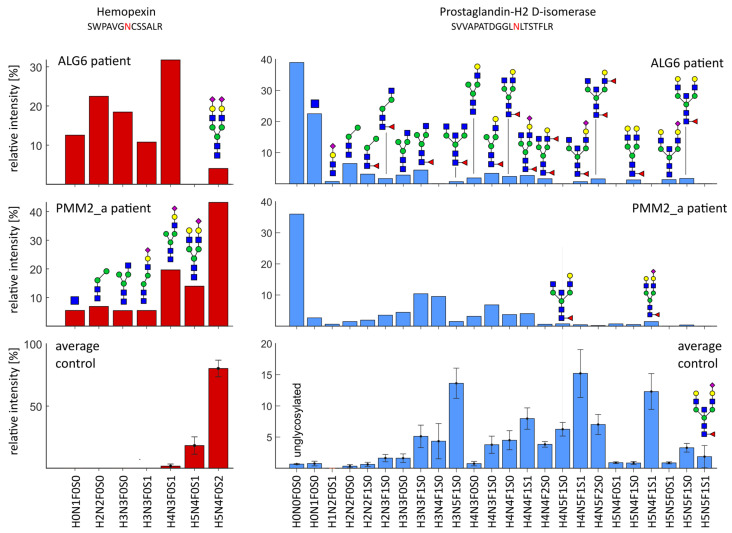
The glycosylation profile of hemopexin and prostaglandin-H2 D-isomerase for the patient diagnosed with ALG6-CDG and PMM2-CDG compared to the control. Both patients show highly truncated glycans on both brain-derived prostaglandin-H2 D-isomerase and non-brain-derived hemopexin. Glycan truncation and a complete loss of glycosylation is more pronounced on brain-derived prostaglandin-H2 D-isomerase, especially seen for the patient diagnosed with PMM2-CDG.

## Data Availability

Output files from MSFragger data processing (glycproteomics processing and proteomics processing) are available as Appendix A. Unprocessed raw data of LC-MS/MS measurements were deposited to the ProteomeXchange Consortium with the dataset identifier PXD038787.

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
