# Peer review of "Glycoproteomics in Cerebrospinal Fluid Reveals Brain-Specific Glycosylation Changes"

_ijms, 2023, doi:10.3390/ijms24031937_

Round 1

Reviewer 1 Report

The author applied a glycoproteomics method to analyze the glycan site occupancy of CSF proteins, also to patients diagnosed with various genetic glycosylation disorders. Via this method, they observed the difference between brain- and blood-derived proteins, defined brain- and blood-type glycan features, and found heavily truncated glycans and complete loss of glycans in brains-derived proteins in patients with severe glycosylation alterations, This discovery further suggests the importance of brain glycosylation in neurological diseases.

Comments:

1.     In Figure 1, the bottom part should be in Figure 1A or figure legend.

2.     The ms/ms spectrum should be provided for the glycan site mentioned in the manuscript, e.g line 128-129

3.     In Figure 2, the name of different glycans should be labeled in the figure.

4.     More evidence should be provided to support that “It is likely that the mixed glycosylation pattern present on a glycoprotein relates to the relative proportion of protein expressed by the brain versus proteins protein coming from the bloodstream and being transported to the CSF via the blood-CSF barrier.” In Line 224-225.

5.     The calculation method for the relative intensity for Figure 3 was missing.

6.     In line 217, for dickkopf-related protein 3, there were four instead of two brain-type glycan motifs as shown in Figure 3.

7.     In Line 222, it should be “ we detected

8.     The upper part for the description of different glycans should also be included in Figure 2.

Author Response

We thank the Reviewer for their careful revision and comments that improved the quality of the manuscript. Please find attached our response to the individual comments made by Reviewer 1.   

Reviewer 2 Report

How can one delineate the difference between the two groups and claim that the difference in glycosylation in these two groups is a result of CDG?

Has the significance of certain types of glycans, eg fucosylation, been demonstrated as a potential marker of fibrosis?

Considering that it is also a question of neurodegenerative diseases, is there any significance in high-mannose glycans?

Why should the importance of alcohol intoxication be emphasized if it is known that the majority of plasma proteins are synthesized in the liver and that alcoholism in itself can lead to changes in glycans at the level of the liver's synthetic function, so it is unlikely that plasma and cerebrospinal fluid can be compared in this case?

Hypothesis and conclusion are missing!

Author Response

We thank the Reviewer for their insightful comments and swift response. Please find attached our response to the individual comments made by Reviewer 2.   

Reviewer 3 Report

The manuscript discusses site-specific glycosylation studies of non-enriched plasma samples from CDG patients. Overall, the manuscript is well-written and easy to read. However, there are a few issues that need to be addressed before it is ready for publication.

First, the lack of error bars in Fig. 3 makes it difficult to determine the significance of the observed differences. It is important to include this information in order to provide a clearer understanding of the data.

Second, Fig. 6 only shows the results for one patient. It would be helpful to see data from multiple patients, or at least provide evidence that the observed difference is larger than the error of measurement.

Minor points to consider include the limited number of patients included in the study, especially the single patient with a history of alcohol abuse. This may not be enough to draw significant conclusions, and this should be noted in the discussion. Additionally, it is surprising that all mono-fucosylated glycans are core fucosylated, while doubly-fucosylated glycans are cone and antennary fucosylated. It is possible that the software was unable to determine the location of fucosylation, and this should be discussed in the manuscript. There are methods available for determining the location of fucosylation, and these should be considered.

Overall, with these changes, I support the publication of the manuscript.

Author Response

We thank the Reviewer for their time and effort to revise our manuscript. We agree with the reviewer’s assessment, in particular on the statistical significance of our data. The reviewer’s comments were helpful and increased the quality of our manuscript. Please find attached our response to the individual comments made by Reviewer 3.   

Round 2

Reviewer 2 Report

Nop